# The Triplex-Centric Assembly and Maturation of the Herpesvirus Procapsid

**DOI:** 10.3390/v17091153

**Published:** 2025-08-22

**Authors:** J. Bernard Heymann

**Affiliations:** National Cryo-EM Program, Cancer Research Technology Program, Frederick National Laboratory for Cancer Research, Leidos Biomedical Research, Inc., Frederick, MD 21701, USA; heymannb@mail.nih.gov

**Keywords:** icosahedral virus, double-stranded DNA virus, antiviral drugs, mature capsid, viral scaffold, capsomer, hexon, penton, triplex, viral portal, cryoEM

## Abstract

Herpesviruses are prevalent infectious agents in humans, with complex structures and life cycles. The viability and detail of a model of capsid assembly and maturation can now be examined against the recently available mature herpesvirus capsids structures. The first large assembly product is the icosahedral procapsid with an outer shell composed of major capsid proteins (MCPs) connected by triplexes (heterotrimers composed of one Tri1 protein and two Tri2 proteins), and an inner shell of scaffold proteins. The asymmetric triplexes have specific and conserved orientations, suggesting a key role in assembly. In the mature capsid structures, triplexes bound to three MCPs may represent an assembly unit where, in most cases, the N-terminus of one MCP wraps around the E-loop of another MCP. The model accommodates the incorporation of a portal into capsid, required for genome encapsidation and viral viability. Cleavage of the scaffold triggers maturation of procapsid. Each of the MCPs rotates mostly as a rigid body, except for the flexible peripheral parts that remodel to close the capsid inner surface. Angularization of the capsid shifts the portal outward to a better contact with the capsid shell. Understanding these events in the herpesvirus life cycle to atomic detail could facilitate the development of drugs that uniquely target assembly and maturation.

## 1. Introduction

Herpesviruses are common pathogens in animals and cause several diseases, ranging in severity from mild to life threatening. Because these viruses are well-integrated parasites of their hosts, they are difficult to manage. Most of the available anti-herpesvirus drugs rely on targeting the viral DNA polymerase, but with the danger of promoting the emergence of resistance [1,2,3]. Herpesviruses are complex entities with elaborate life cycles [4]. For a recent review of the early life cycle see [5]. They have large genomes and are composed of numerous proteins, offering abundant opportunities for the development of drugs [6,7]. At the same time, their complicated structures are still being elucidated.

Seven of the nine human herpesvirus mature capsids have been solved by cryo-electron microscopy (cryoEM) to high enough resolution to fit atomic models into the resultant density maps (Table 1). Furthermore, cryo-electron tomography allowed the visualization of structures not amenable to the averaging required in cryoEM [8,9]. Unfortunately, the procapsid has not been solved to high resolution, and we must interpret changes in conformation based on the available low resolution maps [10,11,12,13]. However, we can relate the corresponding parts to the high resolution structures to identify regions that may be useful as targets for drugs development [7]. Even in the low resolution structures the handedness is evident, with the correct hand established by tilt-pair cryoEM [14]. The hand is particularly noticeable from the orientations of the asymmetric triplexes and confirmed in the high-resolution structures (Table 1).

The complex nature of the capsid and the large conformational changes during maturation present ample opportunities to interfere with the production of viable capsids. Combined with historic data, I explore the molecular interfaces involved in assembly and maturation, and how these can in principle be disrupted. With the plethora of high-resolution structures of the mature capsids (Table 1), it is appropriate to examine the assembly model as it relates to the final structures of the proteins.

## 2. The Architecture of the Herpes Capsid

All the human herpesviruses share the same overall architecture of the capsid and presumably a similar progression of the early stages of its life cycle. The most studied member is the human herpes simplex type 1 virus (HSV1), and the first where the capsid morphology was revealed by negative stain electron microscopy as a T = 16 icosahedral particle [32] and subsequently reconstructed in 3D [33,34]. Its assembly has been studied in some detail *in vivo* [35,36,37,38,39] and *in vitro* [40,41,42,43,44,45]. Assembly requirements have been established *in vivo* for other herpesviruses: CMV [46]; EBV [47]; and KSHV [48]. HSV1 is the only herpesvirus where structural studies have targeted the procapsid and its maturation [11,12,13].

### 2.1. The Genes and Proteins of the Herpes Capsid

The term capsid is used for various compositions of the inner shells of a herpes virion. The first assembled capsid is the procapsid that is composed of the major capsid protein (MCP), the two triplex proteins (Tri1 and Tri2), possibly the small capsid protein (SCP), the portal protein (PP), and the two variants of the scaffold protein (see below for elaboration). The gene and protein names of the proteins incorporated into procapsids are given in Table 2. Procapsids of HSV1 indistinguishable from *in vivo* particles can be assembled from the MCP, the triplex proteins, and the scaffold proteins [40,41,43]. Studies of *in vivo* assembly of beta and gamma herpesviruses using a baculovirus vectors coding for the proteins showed that the SCP is also required for capsid formation [47,48,49,50,51]. The portal protein is not required for assembly of the HSV1 capsid, although it slightly enhances it and must be incorporated to yield capsids capable of packaging DNA [45]. While I concentrate on the minimal proteins needed to assemble procapsids, other proteins may also bind to the procapsid before and during maturation [52].

The scaffold for the herpesviruses consists of a mixture of two proteins with overlapping genes. The long version (MPSP) contains a maturational protease at the N-terminus and a scaffold protein at the C-terminus [37,53,54]. It can cleave itself at two locations, the R site to separate the protease from the scaffold part, and the M site to release the C-terminus and facilitate maturation [55]. The *HSV1* gene UL26 codes for a 635-residue protein containing the two domains with the R site after residue 247 and the M site after 610 [56,57]. The products of the R site cleavage are the 247-residue maturational protease, VP24, and the 388-residue scaffold protein, VP21. The gene *UL26.5* starts in the middle of the *UL26* gene and codes for a protein, VP22a, identical to the 329 C-terminal residues of the UL26 protein. Both VP21 and VP22a participate as scaffold for the assembly of the procapsid and are interchangeable [35,58]. VP21 accounts for only about 10% of the scaffold [57,59,60]. The scaffold proteins of all herpesviruses bind to the MCP at their C-termini, which the maturational protease cleaves at the M site to allow maturation [37,61,62]. In the rest of this paper, I will refer to these two proteins as constituting the scaffold. Unfortunately, no actual structural information has been generated for the scaffold, and we must rely on biochemical and mutational analysis.

### 2.2. The Icosahedral Herpesvirus Capsid

The herpesvirus capsid consists of an outer shell with T = 16 icosahedral symmetry (Figure 1A) and an inner shell composed of the scaffold without any detectable symmetry. The scaffolding proteins form a lipid-like aggregate called the core of the procapsid.

The MCPs are arranged in 150 hexons, and 9–12 pentons at icosahedral five-fold vertices, with the other vertices occupied by 0–3 portal proteins. The capsomers and portal(s) are further connected by 320 triplexes on pseudo-three-fold locations, each with one Tri1 and two Tri2 subunits. The portal protein has twelve-fold symmetry and is located at one five-fold vertex of most capsids (Figure 1B), although some procapsids have no portal, while others may have more than one [13].

An important feature of the capsid is that the triplexes are specifically oriented (Figure 1B), indicating a guiding principle in assembly. Curiously, the periportal triplexes are rotated 120° counterclockwise compared to the peripentonal triplexes in all the herpesvirus structures (HSV1 [16], PRV [23], CMV [26], and EBV [29]). Thus, while each triplex interacts with three MCPs at a quasi-three-fold axis, the interactions are in some way distinguishable and play a role in capsid assembly.

### 2.3. The Scaffold Core

The scaffold has multiple roles in the assembly of the procapsid. Both scaffold proteins attach to the MCPs by their C-termini [37,46,61,62,66]. The scaffold also has a nuclear localization signal (NLS) targeting the MCP-scaffold complex to the nucleus [67,68]. Residues of the HSV1 MCP important for interacting with the scaffold are located at the N-terminus [69,70]. The most prevalent MCP-scaffold complex observed is about 200 kDa, indicating one MCP to 1–2 scaffold molecules [43]. Two-hybrid studies concluded that the scaffold dimerizes, and this dimerization is required for binding to MCP in both HSV1 and CMV [61,62,71]. Incomplete capsids of HSV1 in *in vitro* assembly shows equal amounts of scaffold and capsid, suggesting that the assembly unit includes both [41].

In reconstructions of the HSV1 procapsid, the scaffold is extended from close to the capsid center to the capsid outer shell floor, about 300 Å (Figure 2C) [13,72]. If the scaffold protein (329 residues) and the C-terminal proteolytic product of MPSP that functions as the other scaffolding protein (388 residues) are mostly in coiled coil form, the helical part would account for about 200 residues. In the maps of the procapsid without the maturational protease, the inner part of the scaffold (between radii 180 and 260 Å) is the densest, suggesting some fold more complicated than a coiled coil. The intermediate part (between radii 260 and 420 Å) is less dense and is the most likely part that is coiled coil. The C-terminal part is not seen in reconstructions because it is averaged out. If it is also alpha-helical in nature, it would account for about 40 residues prior to the part bound to the MCP.

### 2.4. The Portal at One or More Capsid Vertices

The first study to quantify the portals in HSV1 capsids was performed using immunolabeling and negative stain EM [73]. The number of portals per capsid is consistently above one: 1.2 ± 0.2 in isolated B-capsids [73], 1.3 ± 0.1 in in vitro assembled capsids [44], 1.1 in procapsids [13], and 1.3 in A-capsids [13]. In a better direct estimation, a cryo-electron tomography detected the portals in 3D maps of individual capsids so that all 12 vertices were sampled, finding most with one portal but with significant numbers of two or none [13]. A similar study for KSHV concluded only one or no portal per capsid, albeit in a very limited number of capsid maps [74]. Because multiple portals may be present on any combination of the 12 vertices, asymmetric high-resolution reconstructions in cryoEM that show the portal on one vertex have portals in any other locations averaged out. The occurrence of multiple portals in capsids indicate that they can be incorporated in already partially assembled capsids.

### 2.5. Overview of the Generation of Herpesvirus Capsids in the Nucleus

The herpesvirus capsid proteins are synthesized by the host machinery in the cytoplasm, imported into the nucleus through the nuclear pore complex, and assembled into an initial particle, the procapsid (Figure 2A–D) [11,13]. Once the procapsid is assembled, the maturational protease cleaves the C-termini of the scaffolding proteins. This releases any tension on the outer shell, which transforms the fragile procapsid into a stable mature capsid (Figure 2E–H). The mature capsid is then decorated with additional proteins and a terminase complex at the portal vertex required for DNA packaging. The scaffolding core is released before or during DNA packaging [75]. The resultant capsid is called a C-capsid, characterized by packaged DNA and with some associated proteins bound to the outside. B-capsids still contain a scaffolding core, while A-capsids are empty. Recently, an empty capsid comparable to the C-capsid but without DNA has been reported, named a D-capsid [17]. Al forms of the mature capsid can contain a portal [17]. Lack of the portal results in B-capsids [76], indicating that the expulsion of the scaffold is somehow linked to DNA packaging. Capsids mostly contain a single portal, although it has been shown that procapsids and A-capsids can contain multiple portals [13]. The C-capsids acquire suitable proteins for transition through the nuclear envelope by envelopment and de-envelopment. An intermediate state within the nuclear envelope is mainly this C-capsid, called the primary enveloped virion (PEV) [77]. Many of the subsequent steps in the assembly of the tegument and envelope are still obscure because of the complexity of the herpesvirus virion.

### 2.6. Cytoplasmic Production of Capsid Subunits and Nuclear Import

Apparently, nuclear import of the viral proteins is rapid and somehow they are protected in some way so that any proteolysis (of the scaffold) and assembly occurs exclusively in the nucleus [54]. Given the ready assembly of HSV1 procapsids *in vitro* [40,41], how is assembly in the cytoplasm prevented? The most likely explanation is that the newly synthesized viral proteins are captured by nuclear import proteins before they can aggregate into larger structures. This requires NLSs, identified in many herpesvirus proteins [67,68,78], but not all [79]. This means that some need to be aided to be transported into the nucleus.

In solution, the HSV1 Tri2 forms a dimer, while Tri1 with Tri2 forms a heterotrimer [42,43], presumably in much the same configuration as in the mature capsids. Because of the large interfaces between the triplex proteins, many changes to the sequence of Tri1 affects formation of the complex, except for some N- and C-terminal deletions [80]. The NLS for the HSV1 Tri1 is located at its N-terminus (^50^PRGSGPRRAAST^61^) and is required to import Tri2 [78,81,82].

The import of the MCP requires the scaffold protein [81], while Tri1 is also able to relocate the MCP [82], forming capsid-like particles in the nucleus [35]. The scaffold C-terminus is necessary for the import of the CMV MCP and cannot be exchanged for that of the HSV1 C-terminus [62]. These proteins could therefore be imported in various complexes, including a triplex with three MCPs and six scaffold proteins as the presumed key complex in the assembly model [12,43].

The portal of HSV1, (*UL6* gene product), is imported into the nucleus by itself, although a specific NLS still needs to be identified [83]. However, its ability to form complexes with the scaffold [44] may allow it to be imported with the scaffold. *In vivo* experiments confirmed that the portal is incorporated into capsids with or without an NLS in the scaffold [68]. Purified portal tends to aggregate [44], suggesting that in the cell it may need a type of chaperone, such as the scaffold or one of the importin proteins [83].

### 2.7. The Maturation of the Procapsid

The procapsid is a labile structure that transforms rapidly into the mature form in the cell nucleus (~40 min [84]). The key event triggering maturation is the protease cutting the last 25 residues of the scaffold proteins, releasing the restraint on the MCP. The maturational protease is required for viral propagation, and only low amounts of virus are produced when absent [53]. The proper disposition of the portal in the mature capsid is also dependent on the protease activity [68]. A mutation in the protease slows down the process [84] to the point that the procapsid can be purified and studied [72]. Several structural studies were conducted to determine the structure of the procapsid [10,85], how it matures [11,12], and the disposition of the portal [13].

In the original maturation study, it was noted that the HSV1 MCPs rotate during maturation [11]. Their overall conformations appeared unchanged at the resolution of those maps, and it was concluded that the MCPs transform mostly as rigid bodies. This leads to the condensation of the MCP turrets (upper parts) in the hexons, adopting local six-fold symmetry (Figure 2). This also results in MCPs moving away from the triplexes and severing some contacts. In the floor (at the inner surface of the capsid), the capsomers extend towards each other, forming continuous interfaces on the inside of the shell. Together with the hexagonal packing of the hexons, the closure of the floor flattens the capsid locally, bringing the MCP turrets together. This pushes the penton outwards and gives the capsid its angular appearance. The icosahedrally symmetrized reconstructions may hide any asymmetric progression of transformation, and the procapsid may sample many different conformational changes, leading to different pathways for maturation.

The kinetics of maturation in HSV1 has been studied by using a temperature-sensitive mutant of the maturational protease [84]. The cleavage of the scaffolding protein occurs within about 70 min and only to a completion of ~50%. However, analysis of plate-forming units on cells sampled at time points after shifting to a permissive temperature show viable virions after only 40 min at a point where ~35% of the scaffold is cleaved. Fluorescently tagged proteins indicated a similar time frame for maturation of PRV capsids *in vivo* [86]. DNA packaging closely follows plaque forming kinetics [87]. With an inactive maturational protease, the spontaneous transformation of the procapsid takes at least two days in vitro [11].

## 3. Procapsid Assembly

### 3.1. The Triplex-Centric Model of Capsid Assembly

In the initial study of assembly intermediates of the capsid of HSV1, the most prevalent MCP-scaffold complex found was about 200 kDa, indicating one MCP to 1–2 scaffold molecules [43]. They proposed a scheme where these MCP-scaffold complexes bind to triplexes, and that the triplex with three MCPs attached form the assembly unit that defines the architecture of the capsid, hereafter referred to as the triplex-MCP_3_ complex (Figure 3B). The sequential addition of the triplex-MCP_3_ complexes then leads to a capsid with the observed architecture [12]. To achieve the pattern of triplex orientations (Figure 1B), the asymmetry of the triplexes (Figure 3A) must play a central role in the sequence of events. This pattern of orientations is determined when pairs of triplex-MCP_3_ complexes form MCP-MCP interfaces (Figure 3C). In our model, a penton vertex (Figure 3D) or a portal vertex (Figure 3E) is first formed, resulting in consistent triplex orientations around each vertex. This is then followed by the addition of further triplex-MCP_3_ complexes units until the shell is closed. It is still difficult to envision how the last complex would insert to close the shell. It is likely that the fragile interactions in the procapsid allows it to “breathe” considerably, and that the final complex locks it into its icosahedral arrangement.

The main interactions in this assembly model are (i) within the triplex-MCP_3_ complexes, (ii) the MCP-MCP interfaces, and (iii) the triplex-MCP_2_ complexes with the portal. The contacts between MCPs in their floor domains have been designated in the first high resolution structural paper of the CMV capsid as “Type I” for the intracapsomer interfaces and “Type II” and “Type III” for the intercapsomer interfaces [24]. The latter two types relate to the dimerization interface between capsomers and the N-terminal in one capsomer interacting with the E-loop in another capsomer.

### 3.2. The Role of the Portal in Capsid Assembly

Studies in HSV1 procapsid assembly showed that the portal must be incorporated early, while it is excluded when added later [45]. However, there is only a slight enhancement of capsid assembly with the portal present. Defective mutants of the portal lead to the accumulation of only B-capsids in the nucleus [88]. The ability of the procapsid to assemble with or without the portal means that the incorporation of the portal depends on its availability, i.e., the relative production and import of portals compared to the triplex-MCP_3_ complexes. The observed number of portals per capsid deviates form a binomial distribution that assumes a uniform likelihood for the appearance of a portal at any of the twelve five-fold vertices for a given average occupancy [13]. This means a single portal per capsid is favored, because it can function as a nucleator [45]. However, it does not exclude the occurrence of none or multiple portals per capsid.

The portal of HSV1 is most probably in its dodecameric state before assembly [44,73,89]. It has disulfide bonds that stabilize the 12-member ring and are required for its formation and DNA encapsidation [90]. Its position in the procapsid away from the outer shell (Figure 2C) suggests that the interaction with the scaffold should be important during assembly. During maturation, the portal shifts ~54 Å outward compared to the ~29 Å of the penton [13], indicating that its interactions with the outer shell are initially very weak. The region in the scaffold required for portal incorporation is around residues 143–151 in SP [91,92], which corresponds to interaction of MPSP with the portal at residues 449–457 [93]. These residues lie before the scaffold dimerization domain of 164–219 [71] and far away from the C-terminus that binds to the MCP. The conclusion is therefore that the portal binds to a dimeric state of the scaffold.

While it is appealing to consider 12 scaffold molecules to associate with a portal (as suggested by the stoichiometry of binding in [44]), electron microscopy of the particles show them to be too disorganized to form a nucleating complex. In our model, the portal associates with a triplex-MCP_2_ complex in the nucleus (because the portal substitutes for one MCP). Importantly, the triplex is oriented 120° rotated relative to that for the penton (Figure 3E), observed in all herpes capsids [16,23,26,28]. It is conceivable that a larger complex (e.g., two triplex-MCP_2_ complexes) could bind to the portal. As for the penton, the portal vertex is then constituted by the addition of the other triplex-MCP_2_ complexes. Because the assembly with or without the portal have similar kinetics [45], the relative formation of the penton and portal vertices is likely driven by the availability of the constituent complexes.

### 3.3. The Triplex-MCP_3_ Complex

The key structure in the assembly model is the triplex-MCP_3_ complex [12,43]. This complex has not been observed on its own and requires some justification. In a previous study, the upper structure of the MCP turret [94] was fitted as a rigid body into the whole series of maturation maps [95]. It is reasonable to assume that the core of the MCP (the P-domain and the turret) mostly retains its conformation during maturation [11]. A similar approach was taken here to fit the triplex-MCP_3_ complex from the mature capsid to the procapsid map, with each subunit moving as a rigid body (using the modeling tools of the Bsoft package [64,65] and ChimeraX [96]). The result is shown in Figure 4A–C for the configuration around the T_b_ triplex in the procapsid, compared to the corresponding structure in the mature capsid (Figure 4D–F). This is not a rigorous fit, and caution should be taken to interpret the results. Nevertheless, the procapsid map imposes some limitations on the orientations of the MCPs. Firstly, the tops of turrets of the MCPs interact in the mature capsid but are well separated in the procapsid, which means that they are much closer in the triplex-MCP3 complex (Figure 4B) and show more interaction with the triplex. Secondly, the separation of the capsomer floors in the procapsid (Figure 2B) means that the triplexes are much more exposed to the inside of the capsid (Figure 2C) as compared to the mature capsid (Figure 2F).

The important feature of the T_b_ triplex-MCP_3_ complex that emerges from the configuration in the mature capsid is that the N-terminus of one MCP, the “N-lasso”, is wrapped around the E-loop of another MCP (Figure 4D). The arrangement of the three N-lasso-E-loop interactions was mentioned in the first structure of the CMV mature capsid, referred to as an “N-lasso triangle” [24]. In the procapsid model of the triplex-MCP3 complex, the rotation of the MCPs move N-termini and E-loops away from each other (Figure 4A). However, the ability of the N-terminus to adopt different conformations, its extended nature in the mature capsid, and the flexibility of the E-loop suggests that they may already interact in the procapsid and, by extension, in the triplex-MCP_3_ complex.

While the core of the 16 different MCPs in the asymmetric unit of the mature capsid adopt the same conformation, there is variation in the peripheral parts: the N-terminus, the E-loop, and the dimerization domain (Figure 5). The N-termini of the penton and P6 MCPs have markedly different conformations compared to the N-lasso of the other 14 MCPs. The P6 N-terminus around the portal is not included before residue 50 for the HSV1 virion [16] and is modeled as a different conformation for the HSV2 portal [19]. We therefore do not have a clear picture of the N-terminus around the five-fold vertices that would determine the configuration around the T_a_ triplex. Nevertheless, they may extend to the other MCP to interact with its E-loop, suggesting that it is still relevant within the triplex-MCP_3_ context.

It is difficult to envision how the N-terminus would wrap around the E-loop after the procapsid is assembled, threading through a very complicated series of cavities. It is more likely that it forms prior to the assembly in the triplex-MCP_3_ complex. This also prepares the interface for the MCP-MCP interactions in the floor that involve the E-loop during later stages of assembly (see below).

### 3.4. The MCP-Scaffold Interaction and the Triplex-MCP_3_ Complex

One of the problems with the triplex-MCP_3_ complex model is that the N-terminus is supposed to be bound to the scaffold during assembly. These regions of the MCP have been predicted as two helices: 22–42 and 58–72 [97]. However, in the mature capsid structure, 38–41 is a ß-strand that associates with the E-loop ß-sheet with other N-terminal residues wrapped around it. The other part (50–70) is the long loop traversing the space between MCPs. The complication with the assays measuring the interaction between the MCP and scaffold is that the results can be interpreted in different ways. The mutations in the N-terminus of MCP may affect the conformation required to bind the scaffold but may not be at the binding site. The pattern of mutations versus activity may reflect the difference between a proper N-terminal fold and one that is unable to bind scaffold. If the triplex-MCP_3_ complex model is correct, it allows for only 10–20 residues at the N-terminus to interact with the scaffold.

### 3.5. Triplex Orientations Define MCP-MCP Interactions

The subsequent steps in the assembly process are the series of associations of pairs of triplex-MCP_3_ complexes (Figure 3C). Because the triplex orientations are consistent within a capsid (Figure 1B) and across all herpes capsids, it must have an important role in assembly. Each pair of triplex-MCP_3_ complexes has two MCP-MCP interacting interfaces (Figure 3C). There are six possible arrangements of such pairs, specified by the opposing triplexes (Figure 6). Excluding the portal vertex, five of the six cases are represented in the capsid. The sixth case (pink box in Figure 6) is the arrangement of the T_a_ triplexes around the portal, with replacement of the MCPs labeled 1 by the portal. This means that two MCPs attached to Tri1 subunits cannot bind, and this arrangement should be considered as unfavorable in capsid assembly.

If we adopt the rule that this unfavored arrangement is completely excluded, we can limit the configurational possibilities. With arbitrary triplex orientations, there are 3^5^ = 243 possible configurations in the asymmetric unit (excluding the T_f_ triplex). According to the rule, the T_f_ triplex can only orient in opposition to the #2 position of the T_e_ triplex, which fixes the latter’s orientation. Because the portal is specifically bound to the Tri1 subunit (see below), the periportal T_a_ triplex is assumed fixed as well. The remaining four triplexes can only adopt four possible configurations (see Appendix A for the three alternate configurations). Additional constraints are needed to select the correct configuration from these four possibilities.

### 3.6. The Intracapsomer MCP-MCP Interfaces in Assembly

Within the capsomers of the mature capsid of all the herpesvirus capsids, the MCPs interact with their neighbors along the whole length from the floor to the top of the turret. In the procapsid of HSV1, the turrets are splayed apart (Figure 2A), indicating that the important interactions for assembly are in the floor and lower part of the turret.

In the HSV1 floor, the E-loop of one MCP fits into a pocket of its neighboring MCP capped by a triplex (Figure 7). The E-loop (residues 104–112) lies on top of the neighboring P-domain, specifically the kink in the spine helix (residue 175) and the ß-sheet (residues 129–131). Above the E-loop, the triplex bound to the neighboring MCP caps off the pocket. The N-terminus of another MCP wraps around the ß-sheet in the E-loop, forming a third strand in the sheet. This conformation may already be formed in the triplex-MCP_3_ unit, stabilizing the E-loop as it slides into the cavity between the adjacent MCP and the triplex.

### 3.7. The MCP-Scaffold Interactions May Also Relate to Asymmetric Triplexes

Multiple studies have now shown that B-capsids contain densities on the inside of the outer shell ascribed to scaffold [17,26,98]. The best representation is in the map of the CMV B-capsid, where there are small densities at nine places [26]. Four pairs of these densities are in locations determined by the orientations of the four triplexes (T_b_-T_e_), while the nineth is located at the T_f_ triplex on the three-fold axis, thus with an ambiguous relationship to the triplex orientation. There were no densities detected associated with the T_a_ triplex, in agreement with the other studies.

The scaffold in the B-capsid may only be partially cleaved, as it was reported that only about half was cleaved during a kinetic study of maturation of the HSV1 capsid [84]. This agrees quantitatively with the nine densities associated with the 16 MCPs in the asymmetric unit found for the CMV B-capsid [26]. Because the five-fold vertices are devoid of these scaffold remnants, it is likely that only the cleavage of scaffold at the five-fold vertices are required to trigger maturation.

### 3.8. The Triplex Interactions with the Portal

At some stage during assembly, the portal as a dodecameric structure interacts with triplexes, either by themselves or in complex with MCPs. If the latter, it must be a triplex-MCP_2_ complex where the missing MCP should have some consequence. The N-terminus of P6 at the portal, as it is at the penton, is in a different conformation as the N-lasso of most of the other MCPs (Figure 5). The N-termini of the interfaces between either pentons and hexons, or portal and hexons, are distinct and may have a specific role in assembly. In the triplex-MCP_2_ complex, at the portal there is only the one N-lasso of P1 wrapped around the E-loop of P6. This is also true for the triplex-MCP_3_ complex at the penton, where the N-termini of the penton and P6 MCPs do not extend to the other MCPs to form the N-lasso. The significance of this will be discussed later regarding an alternative assembly model.

The portal adopts different positions and slightly different conformations in the various types of capsids. The HSV1 dodecamer is composed of two flexibly linked parts, the basket (residues 26–307 and 494–623) and the turret (residues 337–473) [17]. The turret is in a compact conformation in A- and B-capsids but extended in the virion and C-capsids. The B-capsid is the closest to the procapsid and should therefore give the best impression of interactions between the portal and rest of the capsid shell that relates to the procapsid. The turret (or “tentacle”) helices were previously characterized as a leucine zipper (422–443) [88] and were built into the map as such (Figure 8A) [17]. One of these helices contact the T_a_ Tri1 subunit, possible through an ionic bond. In the model, R151 and R152 on the turret helix are close to D257 and E258 in a loop of the Tri1 of the T_a_ triplex. The other contacts are the helix anchors on the portal wing domain with the periportal MCPs in the mature capsids.

The capsomers in the HSV1 procapsid are not in contact with each other (Figure 2B), and this may be true of the portal in terms of the periportal P-hexons (Figure 2C). The assembly therefore progresses through the periportal triplexes and their association with the portal tentacle helices. The remodeling of the MCP floor during maturation brings the N-terminus and dimerization domains into contact with the portal, as seen in a high resolution structure [16]. The symmetry mismatch is accommodated by the flexibility of the linkage between the portal basket and tentacle helices.

The portal–scaffold interaction is more extensive, indicating it is important in assembly [44]. In the procapsid, the portal is embedded in the scaffold (Figure 2C) [13], suggesting that it assembles with an ample complement of scaffold. Extra densities have been observed in A- and B-capsids of HSV1 that were ascribed to scaffold [17].

### 3.9. The Spherical T = 7 Icosahedral Particle Composed of MCP-Tri1

It has been reported that only the full triplex of HSV1 is required for binding the MCP [42]. However, with Tri2 omitted, a smaller particle (~700 Å) is assembled without scaffold [35,82]. Structural analysis showed that it has T = 7 icosahedral symmetry with distorted hexons [99]. There is only one hexon in the asymmetric unit, and two Tri1 molecules plus the one on the three-fold axis (which accounts for a third per asymmetric unit). The first two Tri1 copies have the same orientation as the T_a_ and T_c_ triplexes in the mature capsid. This means that Tri1 has strong enough bonds with the MCPs to favor the same arrangement around the penton. The capsomer floors are well separated, similar to those in the procapsid. This may mean that the full triplexes are required to undergo a similar maturation as observed for the procapsid.

### 3.10. The Small Capsid Protein in Capsid Assembly

In the alpha herpesviruses, the SCP is not needed for assembly, although it is imported with the MCP-scaffold [82]. In the beta and gamma herpesviruses, the SCP is required for assembly: CMV [100]; KSHV [48,49,50]; and EBV [47]. The CMV SCP also appears to be imported into the nucleus already bound to MCP-scaffold complexes [51,82]. These distinctions likely do not impact the nature of our assembly model.

## 4. Procapsid Maturation

### 4.1. Severing the Bond Between Scaffold and MCP

The trigger for maturation is the severing of the MCP-scaffold bond [54]. However, only about half of these bonds are cleaved in a time span of about 40 min [84]. Therefore, it is possible that some of the scaffold protein remains attached to the MCP. Several structural studies attempted to locate scaffold associated with the floor of the capsid. Zhou et al. calculated difference maps between wild type and protease-minus HSV1 capsids, revealing large densities on the inside of the capsid floor [98]. Stevens et al. found relatively large scaffold densities next to the capsid floor except around the 5-fold vertices [17]. Li et al. calculated the difference between A- and B-capsids of CMV, obtaining small densities at nine locations [26] then argued that, while it may be retained on B-capsids, it should be absent on A-capsids.

The 10–20 N-terminal residues of the MCP may allow it to be bound to the scaffold, even when it is also bound to the E-loop of another MCP. The different conformations of the N-termini of the penton MCP and the P6 hexon at the vertex may relate to the observation that only a fraction of the scaffold is cleaved during maturation [84]. The angularization of the capsid is thus a consequence of the cleavage of the scaffold at the five-fold vertex, whereas it is not necessary at the other MCPs and may even be inaccessible there. If there is still scaffold bound to the MCPs, it may have shifted because of the extensive remodeling of the capsid during maturation.

### 4.2. Maturation Involves the Rotation of the MCP Almost as a Rigid Body

The overall shapes of the individual MCPs in the procapsid and mature capsid change little, suggesting rigid body movements during maturation [11]. This is supported by the similarity between the structure of the isolated upper domain of the MCP [94] compared to its structure in the mature capsid [15]. If we fit the mature capsid structures into the procapsid map, two features of the procapsid hexons need to be reproduced. These are the greater separation of the MCP upper domains and the more compact floor domains extending further to the inside of the capsid compared to the mature capsid. The isolated upper domains [94] were originally built into the maturation maps, and the rigid body orientations refined using a Monte Carlo Metropolis (MCM) algorithm [95] (program **bmonte**). The upper domains are well separated in procapsids, and their shapes are distinctive at the low resolutions of the maps [95,101] (Appendix A). More importantly, in the mature capsid maps, the upper domains are readily positioned to minimize clashes.

The upper domain fits were used here to orient the full MCPs from the mature capsid (PDB 6CGR) [15] into the procapsid map (program **bmodfit**) (Figure 9). This was then refined against the procapsid map to alleviate clashes and improve the fit (program **bmonte**). No effort was made to avoid clashes altogether, as I assumed that some flexibility in the structure would accommodate the interactions in the procapsid. The one change that was deemed necessary was the removal of the N-terminus because it intercalates into an MCP of an adjacent capsomer in the mature capsid, extending over the gaps between the capsomers in the procapsid. The first 69 residues were therefore removed prior to performing the fits reported here.

All 16 MCPs in the asymmetric unit show the same trend in the fitted orientations in the procapsid compared to the mature capsid. In the floor, the spine helix that protrudes in the mature capsid is rotated inwards towards the capsomer axis and downwards towards the inside of the capsid (Figure 9). This separates the E-loop of one MCP from the helix and ß-sheet of its neighbor in the capsomer, whereas these parts interact in the mature capsid. If this interaction is also present in the procapsid, the E-loop and N-termini are likely in different conformations. It is perhaps the tension in this interface that is relieved during maturation. These results suggest that the rigidity of the MCP is mostly maintained during maturation, and any potential clashes are mitigated by conformational changes in loops.

### 4.3. Formation of the Intercapsomer MCP-MCP Interfaces During Maturation

The interactions between the capsomers involve much larger changes. The dimerization domains fall outside the procapsid floor in the rigid body fits (Figure 9B), indicating that they only form during maturation. As the MCP rotates from its position in the procapsid, it brings the dimerization domains of neighboring capsomers closer, inducing the conformational change that results in the formation of the two pairs of helices in the hexon–hexon interface and the extended helices in the penton–hexon interface.

The intercapsomer interfaces in the MCP floor domains are between pairs of capsomers, with each interface composed of four MCP subunits. The hexon–hexon interfaces are also different from the hexon–penton interfaces. The hexon–hexon interface in HSV1 (Figure 10A) has an approximate two-fold symmetry with two pairs of interacting helices (314–322 and 329–341) of the dimerization domains from the closest MCPs [15]. The other two MCPs each show two interacting parts, the one with a tyrosine (Y89) in a loop close to a threonine (T151) at the tip of the spine helix of the other MCP, and the second with the N-terminus (N-lasso) that intercalates into the opposing capsomer and wraps around its two-strand sheet (94–100 and 117–123) from the E-loop [15]. As discussed before, the N-lasso seen in 14 of the 16 MCPs in the asymmetric unit may already form during the assembly of the procapsid.

In HSV1, the hexon–penton interface (Figure 10B) features two long helices (329–356) of the closest MCPs that are remodeled versions of the two pairs of short helices in the hexon–hexon interface [15]. The other two MCPs show more extensive interactions of the loop at the end of the spine helix. Their N-termini are folded back on themselves to form short ß-sheets (not the N-lasso fold), and their interaction in the interface is less prominent. This configuration may be the reason the penton is susceptible to extraction by guanidine [60,102] and NEM [103].

There is considerable variation in the dimerization domains of the herpesvirus subfamilies [23,24,25,30]. Thus, we expect that the capsid floor would similarly close in significantly different ways. Unfortunately, we do not have any structural information on procapsids and their maturation other than for HSV1.

### 4.4. Triplex Connection Changes During Maturation

Based on the low-resolution maps, the triplexes do not change much, perhaps just slightly shifting in concert with the movements of the MCPs. The biggest change here is the extension of the floor under the triplexes as the MCPs rotate. Higher up, the connections to the MCP turrets must be adjusted as the capsomers coalesce. Most notable is the space appearing between the triplex and an MCP on the one side, emphasizing its asymmetry [11].

As we only have the mature capsid structure, we need to extrapolate towards the configuration in the procapsid. From the outside view of the procapsid, the most notable connections to the three attached MCPs have been labelled c1–c3 [11]. In the mature capsid, the c2 connection has been severed, with a canyon separating the triplex from the MCP, and a new connection, c4, formed between Tri2 and an MCP (Figure 11A). The c4 connection is problematic because the HSV1 structure has a missing loop here. Speculatively, this loop could extend over to the neighboring MCP, forming the thin connection seen in the low-resolution map.

The triplex connects to the MCP floor, each subunit involving the P-domains of two MCPs from different capsomers, as well as an E-loop from a third MCP (Figure 11B). The E-loop tip is sandwiched between the triplex above and the MCP floors below. This constitutes the pocket as shown in Figure 7. The triplex must therefore be integral to the insertion of the E-loop during assembly of MCP-MCP pairs.

## 5. The Prospects for Drug Development Targeting Assembly and Maturation

Most drugs against herpesviruses aim at inhibiting viral enzymes, with the current standard treatment using acyclovir that inhibits the viral thymidine kinase and DNA polymerase [104]. The problem with that is that the active sites of the enzymes are often similar to those of cellular enzymes, leading to undesirable side effects. Perhaps a better strategy is to target events that are unique to the viruses, such as assembly and maturation pathways [6,7]. This has been explored to some extent but with moderate results and EC_50_ (50% effective concentration) in the range of µM.

### 5.1. Thio-Urea-Based Inhibitors Affect Portal Incorporation

The drug WAY-150138 (also the related CL-253824, EC_50_ 8–20 µM) inhibits the encapsidation of the viral genome in HSV1 (EC_50_ 0.4 µM) and, to a lesser extent, in HSV2, VZV, and CMV [105]. Mutational analysis implicates the *UL6* gene, which codes for the portal protein. Subsequent analysis showed that the drug reduced incorporation of the portal into capsids by inhibiting its association with the scaffold [44,106]. The consequence is an accumulation of B-capsids, mostly without portal, and unable to encapsidate the genome. Mutations that affect the encapsidation were found at the bottom of the portal basket (E121D, A618V, Q621R) [105] where it would interact with the scaffold in the context of the B-capsid (Figure 2C). A slight alteration of the drug switches specificity to the portal of VZV (EC_50_~1 µM), with little activity against HSV1 and CMV [107].

The drug 35B2 (2-[(2,6-dichlorophenyl)methylthio]-3H-pyrazolo [1,5-c]1,3,5-triazin-4-one) reduces VZV plaque formation with an EC_50_ of 3–5 µM [108]. It was characterized as binding to the MCP and preventing capsid assembly. It does not prevent nuclear localization of the MCP but causes localized aggregates of the MCP in the nucleus. Overexpression of scaffold restores the normal nuclear distribution but without forming capsids. Mutations conferring resistant to the drug cluster in the floor of the MCP, suggesting that the drug alters the conformation of the floor to interfere with assembly.

Another drug, 45B5 ([(5-chlorobenzo[b]thiophen-3-yl)methyl] [(4-chloro-phenyl)sulfonyl]amine), inhibits VZV DNA production and suppresses replication in cells with an EC_50_ of 10–20 µM [109]. Resistant mutants map mostly to the portal protein, suggesting that its production is affected by the drug.

### 5.2. Acridone Inhibitor

The compound 5-chloro-1,3-dihydroxyacridone is a moderate inhibitor of HSV1 viral titer with an EC_50_ in the 10–30 µM range [110,111]. The mechanism of inhibition is unclear, while its suppression of the formation of B-capsids suggests an effect on assembly. No resistant mutants have been identified, despite efforts to do so.

### 5.3. Mimicking Peptides as Inhibitors

One idea is to find peptides mimicking parts of the capsid proteins that can enter the cell and compete during capsid assembly and maturation. One region of the HSV1 scaffold (MPSP: ^448^YPYYPGEARGAP^459^) interacts with the portal [93]. A peptide mimicking this region modestly decreases the viral titer with an EC_50_ of about 5 µM [112]. The mechanism of inhibition is ambiguous at best, with some results indicating suppressed protein expression.

Similarly, a peptide mimicking the SCP of KSHV suppresses virion production [30]. The effective concentration could be in the µM range, but the authors only gave the amount used in ng.

## 6. Discussion

It is remarkable that the 1900+ proteins of the herpesvirus capsid assemble into a unique structure. The asymmetry of the triplexes connecting the capsomers have specific orientations that must guide assembly (Figure 1). It is therefore logical to assume a central role of the triplex in determining the sequence of interactions to form a capsid [12,43]. A possible early assembly unit is the triplex-MCP_3_ complex (with scaffold) (Figure 3) that shows N-terminus-E-loop interactions that connect the MCPs (Figure 4). Many possible orientations of the triplex are eliminated by an apparent unfavored interaction between MCPs (Figure 6). The portal presents an alternative nucleation site, with the tentacle helices binding to the Tri1 subunits of the periportal T_a_ triplexes (Figure 8). Maturation of the procapsid proceeds with rotation of the MCPs that allows the closure of the capsid floor, condensation of the hexon turrets (Figure 9), release of some connections between triplexes and MCPs (Figure 11), and outward movement of the portal (Figure 2). Such a complicated assembly scheme offers numerous opportunities for disruption.

### 6.1. Alternative Capsid Assembly Pathways

The triplex-centric model suggests a specific pathway with the triplex-MCP_3_ complex as the first assembly product beyond the MCP-scaffold complex. However, there could be alternate sequences of events that could occur concurrently or preferably. The only way to be sure is to identify these early assembly intermediates by biochemical or structural studies.

One alternative is that the penton may form by itself as a nucleation event, constituting a future vertex like the portal does. Evidence for this is in the N-termini of the penton and P6 MCPs, both not forming the N-lasso conformation that wraps around the E-loop of another MCP in other triplex-MCP_3_ complexes. If the penton forms first, its E-loops are buried so that the N-terminus of P6 cannot wrap around a penton MCP E-loop. Similarly, if the penton MCP N-terminus is already folded in the fully formed penton, it may not extend to the P1 E-loop to form an N-lasso. The consequence is that the next assembly event is the association of triplex-MCP_2_ complexes with the penton, like the expected assembly event at the portal. However, the HSV1 MCPs do not appear to self-associate in solution, with or without scaffolding protein [43]. The only time capsomers have been observed outside of complete or partial capsids has been in the proteolytic dissociation of HSV2 capsids [113].

### 6.2. The Importance of the Scaffold

The scaffold plays several roles in the assembly and maturation of the capsid. It has a nuclear localization signal that facilitates the import of the MCP [81] and the portal [83] into the nucleus. It allows the assembly of the correct size of capsid [114], and its cleavage is the trigger for maturation and the proper disposition of the portal [68]. In the absence of scaffold, partial shells form from the MCP and triplexes [35,36,114]. Particles smaller than the capsid (500–700 Å in diameter) assemble from Tri1 and MCP 500–600 Å [35], some with icosahedral symmetry [99], suggesting that similar forces are relevant in its assembly. The orientations of the Tri1 subunits also appear to correspond to the T_a_ and T_c_ triplexes in the full capsid (Appendix A).

In all cases there are aberrant capsids or shells formed [35], suggesting that there is quite some flexibility in the early assembled structures. This may indicate that the early interactions are weak and allow for extensive trial and error conformational fitting. Only when the correct contacts form, does the capsid start to stabilize. The role of the scaffold is to allow the conformational search process at a radius that is suitable for the assembly of the capsid.

### 6.3. The N-Terminus as a Key Part of Assembly and Maturation

A common theme throughout the structures of the herpes capsids is that the most variable part is the N-terminus of the MCP. Its known roles are (i) binding to the scaffold as a prerequisite for proper assembly and (ii) release from the bulk scaffold by the maturational protease. In the mature capsid, for 14 of the 16 MCPs in the asymmetric unit, it is wrapped around an E-loop of an MCP in a different capsomer, the N-lasso. In the other two MCPs, it adopts a different conformation that lies within the penton–P-hexon or portal–P-hexon interfaces. In the triplex-centric model, the triplex-MCP_2_ complex is presumed to bind to the portal, and the only N-terminus in the N-lasso conformation is that of P1, the only part linking the two bound MCPs. The N-termini of the other two MCPs are thus free to adopt different conformations.

### 6.4. The Organizing Principal of the Triplex

The fixed and consistent orientations of the triplexes over all the herpesvirus capsids indicate a strong organizing principle that guides assembly. For the portal, it appears clear that the interaction is limited to Tri1, excluding the two Tri2 subunits. The orientation of the triplex bound to the portal is excluded from the pairs of triplex-MCP_3_ complexes (Figure 6). The consequence is that there is no interaction between two MCPs bound to Tri1. The conclusion is that the configuration of the MCP bound to Tri1 is sufficiently different from those bound to the two Tri2s.

### 6.5. Do Disulfides Play a Role in Assembly?

It has been known for a long time that reducing disulfides in HSV1 capsids destabilizes them [115]. In vitro, dithiothreitol prevents HSV1 capsid assembly entirely [40]. In HSV1, the scaffold may be cross-linked by disulfides [116]. Studies comparing HSV2 capsid composition under reducing and non-reducing conditions indicated disulfide bonds between the MCP, VP19, and scaffold [117]. HSV1 capsid isolated in the presence of NEM, a compound blocking disulfide formation, resulted in capsids with destabilized pentons and Ta triplexes [103].

In most high-resolution structures of herpes capsids, disulfides are not evident. The exception is the capsid of HSV2, featuring disulfides internal in the triplexes and between Tri1 and the MCP [18]. It is therefore an unresolved issue as to what structural role disulfides play during assembly and the stabilization of the capsid.

### 6.6. The Similarities to Bacteriophage HK97 Assembly and Maturation

The floor of the MCP in herpesvirus capsids has a fold reminiscent of the HK97-fold or Johnson fold in bacteriophage HK97. Even though this association is often mentioned, the details for each virus are very specific, and the comparisons only reveal general trends [118]. The capsid protein of the HK97 bacteriophage is also able to assemble into capsids with or without its portal [119]. The scaffold is the N-terminal 102 residues cleaved off to allow the capsid to mature through its first step from prohead I to prohead II [120], reminiscent of the cleavage of the scaffold in herpesviruses. Coiled-coil helices of the scaffold mediate the symmetry mismatch at the portal [121], while, in the herpesviruses, the T_a_ triplexes bind to the coiled-coils of the portal tentacles.

The E-loop in the Johnson fold emerges as a key structural feature in assembly. In the HK97 phage, the E-loop protrudes from the surface in the prohead and maturational intermediates [122]. While the rest of the protein undergoes rigid body movement, the E-loop changes conformation to allow crosslinking to the P-domain of a distant subunit [123,124,125]. This reflects the interactions of the herpes MCP E-loop with two other MCPs and its likely conformational changes during assembly and maturation. The transient interactions in the assembly and maturation of HK97 supports an induced-fit model [126].

Rigid body rotations with remodeling of the connecting interfaces appears to be the common thread for virus capsid maturation, as for bacteriophage phi6 [127], HK97 [124], and HSV1 [11,95,101].

## 7. Conclusions

The triplex centric model of herpes capsid assembly is supported by several lines of evidence, such as the presence of the triplex-MCP_3_ complex as a unit in the mature capsid and the specific and conserved orientations of the triplexes. It may only be part of the assembly process, as other interactions with different complexes are possible.

One note of caution: the resolutions achieved in the cryoEM maps of the herpesvirus capsids reflect the overall quality, with some parts of the maps better than others. Because of their large sizes, the reconstruction of these capsids is demanding, and the overall details are on the edge of distinguishing atoms. As is often observed, the most interesting parts of the maps may be of lower resolution, and the modeling could be ambiguous. Therefore, the conclusion reached here must be tested against hopefully better structures in the future or targeted mutational studies that could clarify the functionality of parts of the structure.

Nevertheless, the model presents testable features, such as the existence of triplex-MCP-scaffold complexes, the structures of partial assemblies, and the point at which the portal is incorporated. The methods of choice would be cryoEM and cryo-electron tomography of in vitro capsid assembly at early time points, augmented by biochemical and mutational analysis. Ultimately, better understanding the detail of assembly and maturation provides an improved basis for developing measures against herpesvirus infections.

## Figures and Tables

**Figure 1 viruses-17-01153-f001:**
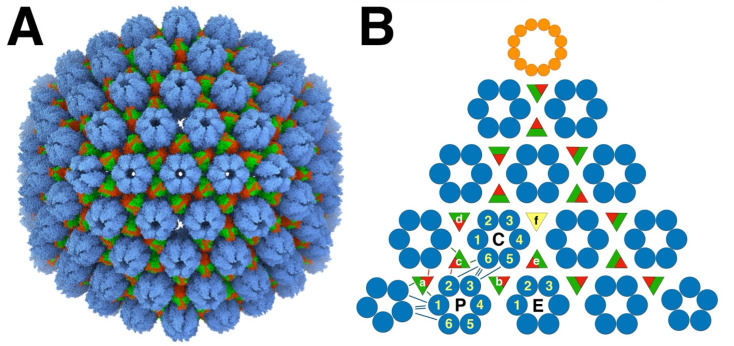
The herpes simplex virus type 1 (HSV1) capsid architecture. (**A**) The major capsid proteins (MCPs: blue) form capsomers, the pentons and hexons, connected by triplexes (red/green). The triplexes are heterotrimeric, composed of one Tri1 subunit (red) and two Tri2 subunits (green). The structure is from Dai and Zhou [15] (PDB 6CGR). (**B**) The layout of one T = 16 icosahedral face of the capsid shows the three hexons in the asymmetric unit: P—peripentonal; E—equatorial; C—central. The different triplexes (a–f) have specific orientations indicated by the red Tri1 subunits. The triplex on the three-fold axis (f) is missing in (**A**) and of indeterminate orientation in (**B**) because the three-fold symmetrization obscures its high-resolution structure. Also shown is the dodecameric portal (orange) occupying one icosahedral vertex. The blue lines indicate the main connections between the capsomers, and the green/red lines the connections between triplexes and MCPs. The numbering of the MCPs (1-6) is according to that established by Yu et al. [24]. (All molecular visualizations were performed with UCSF Chimera X [63] Manipulation of molecular structures to extract individual proteins for composition and fitting was performed with the modeling tools of Bsoft [64,65]).

**Figure 2 viruses-17-01153-f002:**
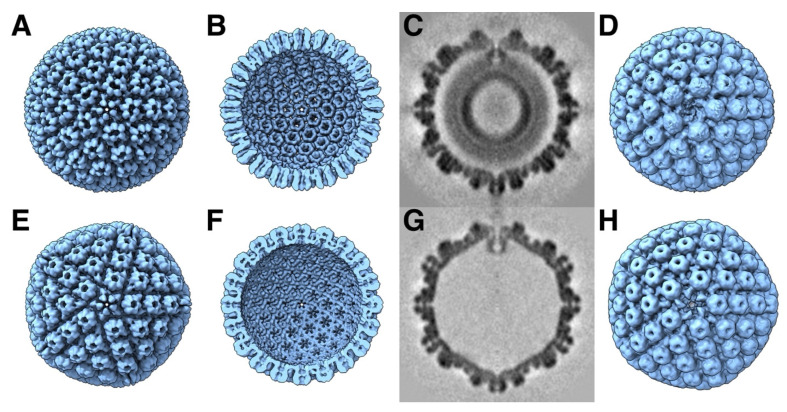
Comparison of the procapsid (**A**–**D**) and mature capsid (**E**–**H**) of HSV1. The main external feature of the icosahedral procapsid (**A**) is its spherical shape with distorted hexon protrusions. The mature capsid (**E**) appears angular with hexons showing regular local six-fold symmetry. On the inside of the procapsid (**B**), the capsomers are well separated, while, in the mature capsid (**F**), the floor is closed between capsomers, with perforations on the five-fold and local three-fold and six-fold axes. An asymmetric subtomogram average of the procapsid (**C**,**D**) reveals the portal at the top vertex interacting with the internal scaffold, replacing a penton at a five-fold vertex. A corresponding asymmetric subtomogram average of the mature A-capsid (**G**,**H**) shows the portal in a position closer to the capsid shell. Maps in (**A**,**B**,**E**,**F**) from [11] and (**C**,**D**,**G**,**H**) from [13] (EMD 22378 and 22379).

**Figure 3 viruses-17-01153-f003:**
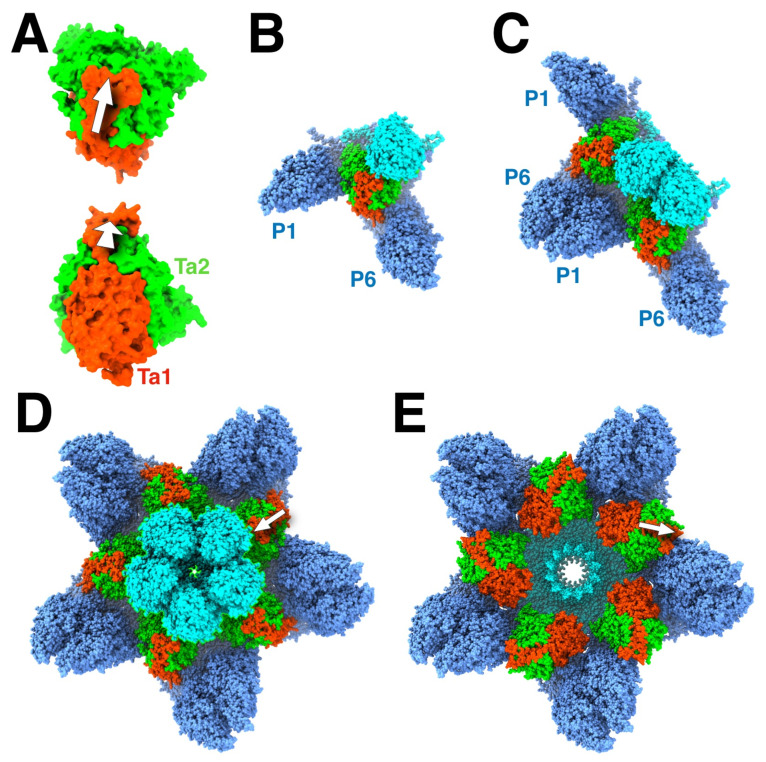
The role of the triplex in assembly initiation of HSV1. (**A**) The triplex from above (**top**) and the side (**bottom**) show the Tri1 subunit (red) with an extending an arm (arrow) over the two Tri2 subunits (green). (**B**) A model of the triplex-MCP_3_ complex, with the light blue indicating the future pentonal MCP, and the dark blue the future P-hexon MCPs. (**C**) The association of two triplex-MCP_3_ complexes, with two MCP-MCP interfaces. (**D**) A model of the assembled pentonal vertex with five pairs of MCPs that will eventually form part of the P-hexons. (**E**) A model of the assembled portal vertex, with its triplexes rotated 120° compared to those from the pentonal vertex (arrows). Panels A-D were assembled from selected chains of 6CGR [15], and Panel E from selected chains of 6ODM and 6OD7 [16].

**Figure 4 viruses-17-01153-f004:**
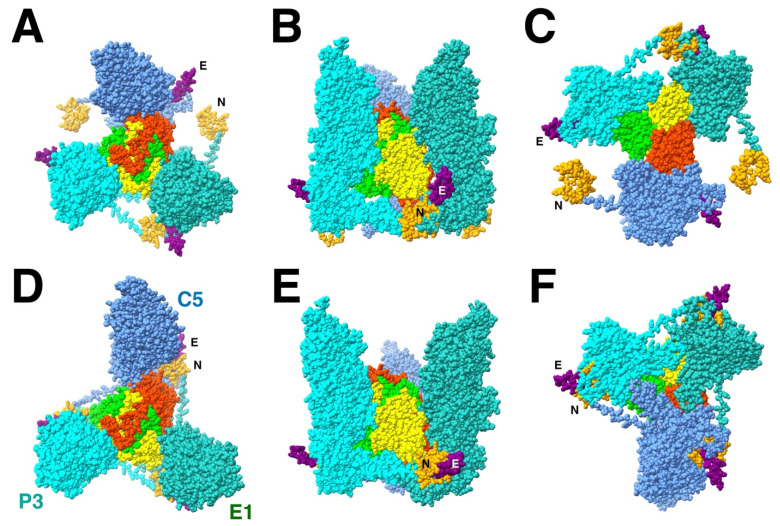
The triplex-MCP_3_ complex for triplex T_b_ (red, lime, yellow) built based on the rigid body orientations of the three MCPs (cyan, blue, green) in the procapsid ((**A**): top; (**B**): side; (**C**): bottom) compared to the corresponding complex extracted from the mature capsid ((**D**): top; (**E**): side; (**F**): bottom). The N-terminus (N: orange) is wrapped around the E-loop (E: purple) in the mature capsid (**D**–**F**) but well separated in the procapsid model (**A**–**C**). The turrets of the three MCPs in the procapsid are much closer together in the procapsid (**B**) than in the mature capsid (**E**). All subunit models from PDB 6CGR [15].

**Figure 5 viruses-17-01153-f005:**
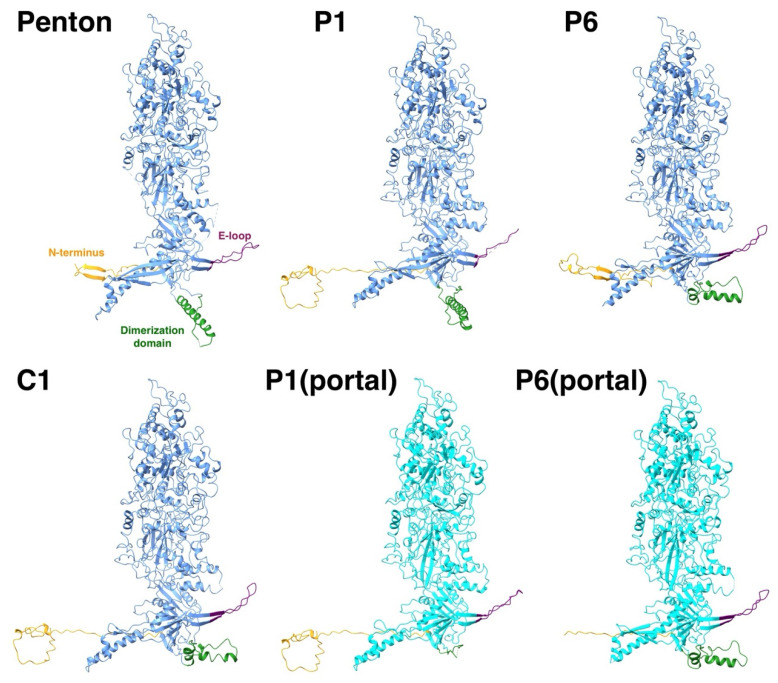
The flexible parts of an MCP are the N-terminus (orange), the E-loop (purple), and the dimerization domain (green). While thirteen of the MCPs in the asymmetric unit adopt the same conformation (e.g., C1), the flexible parts of the MCPs around the penton and the portal have very different conformations (light blue).

**Figure 6 viruses-17-01153-f006:**
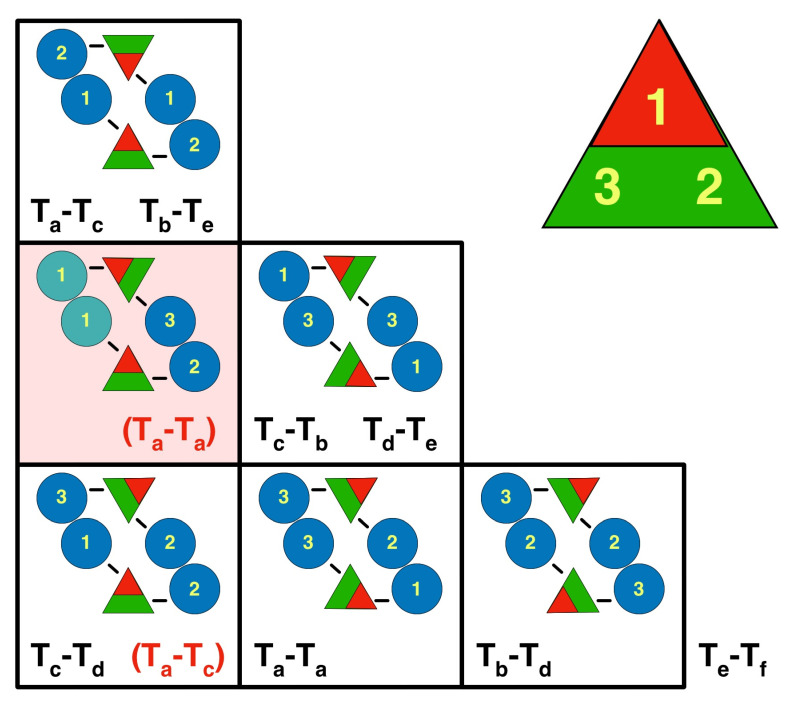
The orientations of the triplexes relative to each other define the pairwise interactions of attached MCPs. The eight pairs of triplexes (black text) and the two alternative pairs associated with the portal (red bracketed text) can be arranged in six possible configurations. The yellow numbers refer to the subunits in the triplex, ordered clockwise around the triplex (1 = Tri1; 2 = Tri2a; 3 = Tri2b). The blue discs indicate the MCPs with the numbers specifying to which part of the triplex it is bound. Because the T_f_ triplex lies on a symmetry axis, it adopts all three the configurations in the bottom row relative to three T_e_ triplexes. The square with the pink background shows an arrangement that never occurs, except around the portal for the T_a_ triplexes, with the portal replacing the MCPs (green) on the left side. The other arrangement around the portal is for T_a_-T_c_ (in brackets in the lower left box).

**Figure 7 viruses-17-01153-f007:**
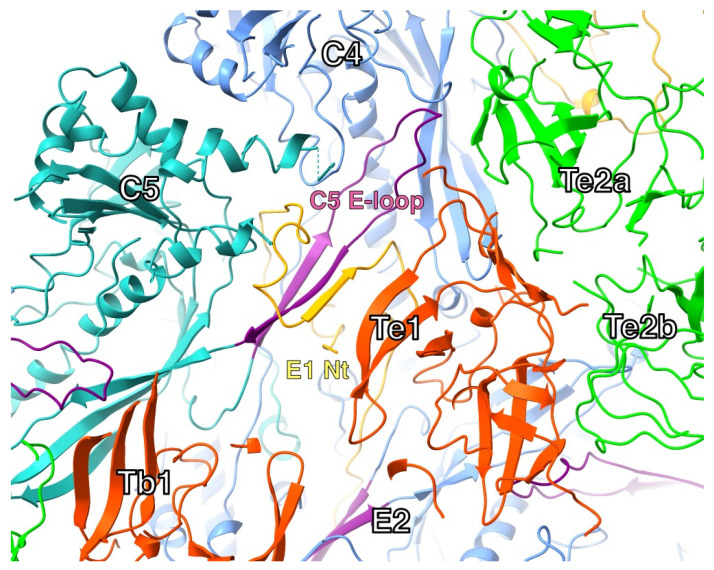
The N-terminus and E-loop in one configuration of triplexes in the mature capsid (T_b_-T_e_). The N-terminus (orange) of one MCP (E1) wraps around the ß-sheet of the E-loop (purple) in another MCP (C5), adding a strand to the sheet. The tip of the E-loop sits in a pocket formed by the adjacent MCP (C4) and the interface between two triplexes (Te1 and Te2a). Model from PDB 6CGR [15].

**Figure 8 viruses-17-01153-f008:**
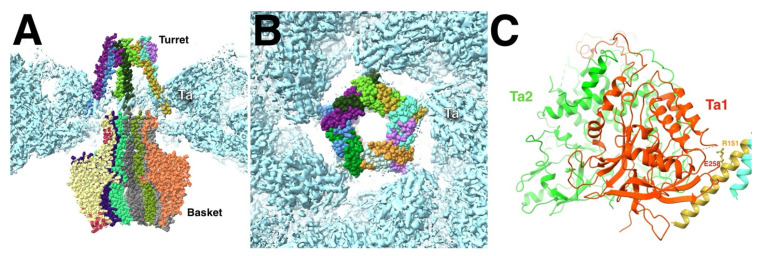
(**A**) The dodecameric portal of the HSV1 B-capsid features five pairs of tentacle/turret helices that interact with the surrounding T_a_ triplexes. (**B**) The tentacle helices only interact at the corners of the pentameric arrangement. (**C**) The acidic residues D257-E258 of the Ta1 subunit are juxtaposed across from the basic residues R151-R152 of the turret helix. Portal vertex reconstruction from EMD-70684 and the portal model from PDB 9OP5,9OPV [17]. Triplex model from PDB 6CGR [15] fitted into the portal vertex map.

**Figure 9 viruses-17-01153-f009:**
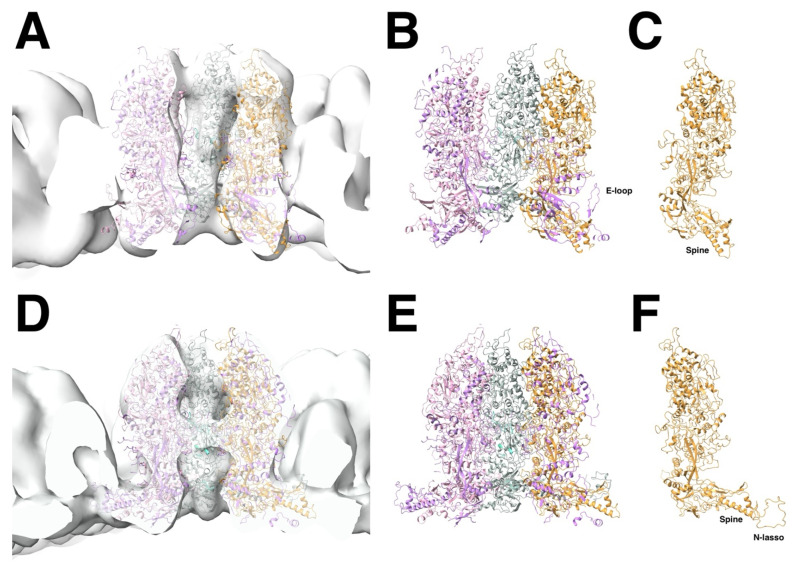
Changes of the E-hexon during maturation. (**A**–**C**) Fit of the E-hexon MCPs, each as a rigid body, into the density of the procapsid (**A**–**C**), compared to the disposition of the mature E-hexon in the density of the mature capsid (**D**,**E**). In the fit to the procapsid, the E-loop points into an unlikely direction that would clash with the triplex. The isolated MCP in the procapsid fit (**C**) shows considerable rotation compared to the MCP in the mature capsid (**F**). The N-terminus (70 residues) was omitted in the fit to the procapsid as it severely protrudes from the density. Structures from PDB 6CGR [15].

**Figure 10 viruses-17-01153-f010:**
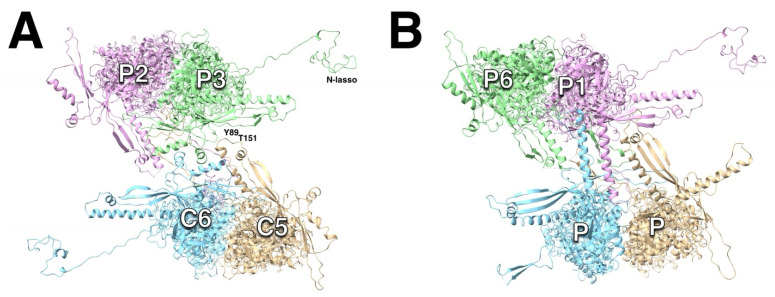
The dimeric interactions between capsomers of HSV1 (PDB 6CGR). (**A**) The hexon–hexon interfaces are very similar in all herpesviruses. Two pairs of helices of the dimerization domains closest MCPs form a local two-fold axis, and the P-domains and N-termini (N-lassos) of the other two MCPs interact. (**B**) The hexon–penton interfaces vary considerably between herpesviruses. In HSV1, the dimerization domains of the closest MCPs form two long helices, while the P-domains of the other two MCPs interact.

**Figure 11 viruses-17-01153-f011:**
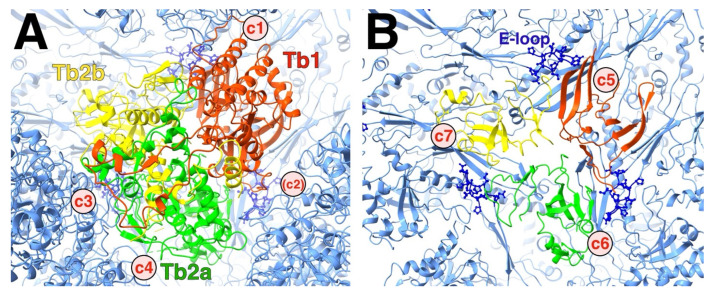
The asymmetry of the triplex. (**A**) Top view of the HSV1 T_b_ triplex, showing the three subunits (red, green, and yellow) surrounded by the connected capsomers (blue). The triplex connections away from the capsid floor denoted c1 and c3 both involve Tri1 (red). The c2 connection that exists in the procapsid is a canyon in the mature capsid. (**B**) The same view as in A but clipped to show the bottom of the triplex at the quasi-three-fold. Each subunit of the triplex sits on top of the interface between two MCP P-domains, constituted of the long spine helix and a ß-sheet. The rough locations of the triplex-MCP connections are labelled c5–c7. The E-loop of an adjacent MCP (dark blue ball-and-stick) fits between the triplex above it and the P-domain underneath it.

**Table 1 viruses-17-01153-t001:** Herpesvirus mature capsid structures.

Herpesvirus Map	PDB Code	EMDB Code	Resolution (Å)	References
Alphaherpesvirinae				
HSV1 C-capsid/tegument	6CGR	7472	4.2	[15]
HSV1 virion portal vertex	6ODM, 6OD7	9860	4.3	[16]
HSV1 A-capsid portal vertex	9OP4	70678, 70683	3.6	[17]
HSV1 B-capsid portal vertex	9OP5	70679, 70684, 70688	3.5	[17]
HSV1 C-capsid portal vertex	9OPC, 9OPV	70725	4.1	[17]
HSV1 D-capsid portal vertex	9OPB, 9OP8	70682, 70687	6.6	[17]
HSV2 B-capsid	5ZAP	6907	3.1	[18]
HSV2 B-capsid portal vertex	6M6I	30125	4.1	[19]
VZV A-capsid	7BW6	30228	3.7	[20]
VZV A-capsid	6LGL	0880	4.3	[21]
VZV C-capsid	6LGN	0881	5.3	[21]
VZV B-capsid ^†^	8XA *	3819 *	5.3	[22]
VZV C-capsid ^†^	8X9 *	3818 *	5.0	[22]
PRV C-capsid ^†^	7FJ1	31611	4.4	[23]
PRV A-capsid ^†^	7FJ3	31612	4.5	[23]
**Betaherpesvirinae**				
CMV C-capsid/tegument	5VKU	8703	3.9	[24]
CMV (murine) virion	6NHJ	9366	5.0	[25]
CMV A-capsid ^†^	8HEU, 8HEV	34698	3.9	[26]
CMV B-capsid ^†^	8HEX, 8HEY	34699	3.7	[26]
HV6B virion	6Q1F	20557	9.0	[27]
**Gammaherpesvirinae**				
EBV virion ^†^	6W19	21504	5.5	[28]
EBV virion ^†^	7BSI	30162	4.1	[29]
KSHV virion	6B43	7047	4.2	[30]
HSHV virion ^†^	6PPI,6PPB	20430	7.6	[31]

* Structures of parts of the capsid are available. ^†^ Maps with portals.

**Table 2 viruses-17-01153-t002:** Genes and proteins of the human herpesvirus capsids.

Protein	Alphaherpesvirinae			Betaherpesvirinae				Gammaherpesvirinae	
	HSV1 HHV1	HSV2 HHV2	VZV HHV3	CMV HHV5	HV6A HHV6A	HV6B HHV6B	HV7 HHV7	EBV HHV4	KSHV HHV8
Outer shell—capsid									
MCP	UL19/VP5	UL19	ORF40	UL86	U57	U57	U57	BcLF1	ORF25
Tri1	UL38/VP19C	UL38	ORF20	UL46	U29	U29	U29	BORF1	ORF62
Tri2	UL18/VP23	UL18	ORF41	UL85	U56	U56	U56	BDLF1	ORF26
SCP	UL35/VP26	UL35	ORF23	UL48/9	U32	U32	U32	BFRF3	ORF65
PP	UL6	UL6	ORF54	UL104	U76	U76	U76	BBRF1	ORF43
Inner shell—scaffold									
MPSP	UL26/ VP21+VP24	UL26	ORF33	UL80/ ACpra	U53	U53	U53	BVRF2	ORF17
SP	UL26.5/VP22a (ICP35)	UL26.5	ORF33.5	UL80.5/ pAP	U53.5	U53.5	U53.5	BdRF1	ORF17.5

MCP: major capsid protein; Tri1 and Tri2: triplex proteins; PP: portal protein; MPSP: maturational protease and scaffold protein; SP: scaffold protein; SCP: smallest capsid protein. (Protein names obtained from https://www.uniprot.org, accessed on 15 July 2025)

## Data Availability

All data are available in the public databases cited.

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
