# Peer review of "The Triplex-Centric Assembly and Maturation of the Herpesvirus Procapsid"

_viruses, 2025, doi:10.3390/v17091153_

Round 1
Reviewer 1 Report
Comments and Suggestions for Authors
Manuscript written by J Bernard Heymann represent a review of current results in a field of capsid assembly in viruses representing Herpes viridae family. Major assays used to get data is cryoEM. The first large assembly product is the icosahedral procapsid with an outer shell composed of
major capsid proteins (MCPs) connected by triplexes, and an inner shell of scaffold proteins. The asymmetric triplexes play a key role in assembly. In the mature capsid structures, triplexes bound to three major capsid protein (MCPs) and represent an assembly unit. The N-terminus of one MCP wraps around the E-loop of another MCP. The model accommodates the incorporation of a portal into capsid, required for genome encapsidation and viral viability. Described data provide a basis for development a novel highly specific anti-viral compounds that target assembly and maturation. Manuscript is very well written, complicated mechanism of viral capsid formation and maturation is presented clear and all text is based on good quality of cryoEM figures. Supplemented figures are good quality.
I have no other comments and recommend to accept manuscript in present form.
Author Response
Manuscript written by J Bernard Heymann represent a review of current results in a field of capsid assembly in viruses representing Herpes viridae family. Major assays used to get data is cryoEM. The first large assembly product is the icosahedral procapsid with an outer shell composed of major capsid proteins (MCPs) connected by triplexes, and an inner shell of scaffold proteins. The asymmetric triplexes play a key role in assembly. In the mature capsid structures, triplexes bound to three major capsid protein (MCPs) and represent an assembly unit. The N-terminus of one MCP wraps around the E-loop of another MCP. The model accommodates the incorporation of a portal into capsid, required for genome encapsidation and viral viability. Described data provide a basis for development a novel highly specific anti-viral compounds that target assembly and maturation. Manuscript is very well written, complicated mechanism of viral capsid formation and maturation is presented clear and all text is based on good quality of cryoEM figures. Supplemented figures are good quality.
I have no other comments and recommend to accept manuscript in present form.
Thank you for the positive comments.
Reviewer 2 Report
Comments and Suggestions for Authors
Heymann is one of the leaders worldwide in the electron microscopy analysis of very large viruses that infect humans. Here, he summarises what is known about the formation, structure and biology of the herpesvirus procapsid. In many ways he is the ideal person to describe the work that he and his laboratory have done to elucidate the structural interactions of these major pathogens. As he says in the Discussion (p24), in this system a capsid assembles containing >1900 proteins. One wonders why such viruses are so molecularly complex, given that other virions make do with many fewer components, yet can be lethal in humans and seem to thrive in the cellular environments created by their hosts. Since we, and other animals, are the hosts used by Herpesviruses in their lifecycles, and since they are known causative agents of life-threatening diseases in humans it is appropriate to study them. However, the revolution that has occurred in many simpler viruses, due to the use of asymmetric cryo-electron microscopy, is largely abrogated in these systems because they are inherently so large. EM images are necessarily dependent on the molecular ordering within complexes being viewed. Herspesviruses are sadly so large that dynamic areas are imaged less well than other areas. These caveats are perhaps under-appreciated in a field where simpler systems are being dissected pretty thoroughly. The result here is a series of necessary speculations that are appropriate but would almost certainly be disallowed in simpler systems. There is an idea, common in virology, that to be infectious virions must by dynamic and if that is the case the most important aspects of herpes virion architecture are not well-resolved, even by the best modern instrumentation. Nevertheless, here he makes the case that Herpes capsid formation is driven by molecular interactions between the triplex of proteins that are seen in the Herpesvirus structures. Sadly, this remains a speculation but a good one.
I would support publication of this Review of work on a complex virion in Viruses. The author makes an argument that his proposal is testable but argues for a structural approach (cryo-electron tomography). In my opinion a combination of structural and functional (genetic) studies would be better.
Heymann is one of the leading virologists worldwide studying complex virions. He is somewhat isolated and has a tendency to write (in English) as he thinks, without appreciating that the people he is writing for are not as familiar with his system as he is. The following is a selection of issues that could be problematic for a wider readership.
Issues to tidy up:
In the Abstract: line 13, please define triplexes at the earliest opportunity (of what?).
Line 20: please define “capsid “floor”.
Line 22: please replace “will” with “could”
Introduction: Line 29 “cause several….”
Line 30: define what is meant by “well-integrtaed”
Line 31: insert surely “anti-herpes virus” drugs….
Line 33: Reference Zarrouk, 2017, surely in error that this appears here, is it meant to be Ref#2?
Line 34: surely “life-cycles” should be hyphenated
Line 35: please define or reference the term “attack surface”
Line 40: …fit atomic models “into the resultant electron density map ..”
Lines 42 & 43: …the procapsid has not “yet” been…” “high-resolution ….interpret changes [to what?]…
Table 2: “please give sources etc for the data”
Line 727: “implicates” not implicated
Lines 895-899: “list more ways to test these features” why choose cryo-electron tomography?”

Slight improvements are suggested at the end of my comments.
Reviewer 3 Report
Comments and Suggestions for Authors
Review of Manuscript “The triplex-centric assembly and maturation of the herpesvirus procapsid” by J Bernard Heymann.
Based on own published data and a comprehensive compilation of recently available structures, the author presents and validates the triplex-centric model of herpesvirus capsid assembly and maturation. The manuscript is well-written and nicely structured. All the individual steps of the capsid assembly and maturation with the function of the individual proteins and structures like the major capsid proteins (MCPs), the triplexes, the portal and the scaffold are explained in detail in a very understandable manner. The corresponding illustrations are excellent and the main text guides very well through these illustrations. The discussion section may be shortened a bit, since it contains some repetitions of the preceding sections. In summary, this is an excellent paper for those interested in viral assembly. A few typewriting errors as listed below should be corrected in a revised version of the manuscript.
Minor points:
1) Line 29, typo: change “causes“ to “cause“ (plural herpesviruses)
2) Line 33, format citation
3) Line 97: Should probably read “scaffolding proteins“
4) Line 860: Change “is“ to “are“ (plural)
Author Response
Based on own published data and a comprehensive compilation of recently available structures, the author presents and validates the triplex-centric model of herpesvirus capsid assembly and maturation. The manuscript is well-written and nicely structured. All the individual steps of the capsid assembly and maturation with the function of the individual proteins and structures like the major capsid proteins (MCPs), the triplexes, the portal and the scaffold are explained in detail in a very understandable manner. The corresponding illustrations are excellent and the main text guides very well through these illustrations. The discussion section may be shortened a bit, since it contains some repetitions of the preceding sections. In summary, this is an excellent paper for those interested in viral assembly. A few typewriting errors as listed below should be corrected in a revised version of the manuscript.
Thank you for the positive comments.
Minor points:
1) Line 29, typo: change “causes“ to “cause“ (plural herpesviruses)
Changed.
2) Line 33, format citation
Fixed.
3) Line 97: Should probably read “scaffolding proteins“
Replaced with "scaffold".
4) Line 860: Change “is“ to “are“ (plural)
Changed.